# Detecting Minor Symptoms of Parkinson’s Disease in the Wild Using Bi-LSTM with Attention Mechanism

**DOI:** 10.3390/s23187850

**Published:** 2023-09-13

**Authors:** Vasileios Skaramagkas, Iro Boura, Cleanthi Spanaki, Emilia Michou, Georgios Karamanis, Zinovia Kefalopoulou, Manolis Tsiknakis

**Affiliations:** 1Institute of Computer Science, Foundation for Research and Technology Hellas (FORTH), GR-700 13 Heraklion, Greece; tsiknaki@ics.forth.gr; 2Department of Electrical and Computer Engineering, Hellenic Mediterranean University, GR-710 04 Heraklion, Greece; 3School of Medicine, University of Crete, GR-710 03 Heraklion, Greece; boura.iro@gmail.com (I.B.); kliospanaki@gmail.com (C.S.); 4Department of Basic and Clinical Neuroscience, Institute of Psychiatry, Psychology and Neuroscience, King’s College London, London WC2R 2LS, UK; 5Department of Neurology, University Hospital of Heraklion, GR-715 00 Heraklion, Greece; 6School of Health Rehabilitation Sciences, Department of Speech and Language Therapy, University of Patras, GR-265 04 Patras, Greece; emiliamichou@upatras.gr; 7Department of Neurology, Patras University Hospital, GR-264 04 Patras, Greece; karamanis6eor@gmail.com (G.K.); zkefalopoulou@gmail.com (Z.K.)

**Keywords:** Parkinson’s disease, Bi-LSTM with attention, in the wild detection, accelerometer, deep learning

## Abstract

Parkinson’s disease (PD) is a neurodegenerative disorder characterized by motor and nonmotor impairment with various implications on patients’ quality of life. Since currently available therapies are only symptomatic, identifying individuals with prodromal, preclinical, or early-stage PD is crucial, as they would be ideal candidates for future disease-modifying therapies. Our analysis aims to develop a robust model for accurate PD detection using accelerometer data collected from PD and non-PD individuals with mild or no tremor during phone conversations. An open-access dataset comprising accelerometer recordings from 22 PD patients and 11 healthy controls (HCs) was utilized. The data were preprocessed to extract relevant time-, frequency-, and energy-related features, and a bidirectional long short-term memory (Bi-LSTM) model with attention mechanism was employed for classification. The performance of the model was evaluated using fivefold cross-validation, and metrics of accuracy, precision, recall, specificity, and f1-score were computed. The proposed model demonstrated high accuracy (98%), precision (99%), recall (98%), specificity (96%), and f1-score (98%) in accurately distinguishing PD patients from HCs. Our findings indicate that the proposed model outperforms existing approaches and holds promise for detection of PD with subtle symptoms, like tremor, in the wild. Such symptoms can present in the early or even prodromal stage of the disease, and appropriate mobile-based applications may be a practical tool in real-life settings to alert individuals at risk to seek medical assistance or give patients feedback in monitoring their symptoms.

## 1. Introduction

Parkinson’s disease (PD) is the second most common neurodegenerative condition, estimated to currently affect more than 8.5 million people globally, according to the World Health Organization (WHO) [1]. An elevated risk for PD driven by population aging and industrialization-related environmental factors is likely to result in a PD pandemic within the following years [2]. PD has been associated with significantly poorer health-related quality of life, even in the early stage of the disease [3,4]. Interestingly, it was found to be the most rapidly growing disease regarding figures of prevalence, disability, and fatality among neurological disorders, with the latter being the primary source of disability worldwide [5]. Consequently, and since its global burden is expected to surge in the future, interventions for early diagnosis and efficient treatment are imperative.

Care management for PD is personalized and multifactorial in nature [6]. Early recognition of symptoms allows a timely and informed introduction to efficient multilevel/multidisciplinary approaches, both pharmacologic and nonpharmacologic (e.g., exercise, speech/occupational therapy), as clinically indicated, with a direct and significant impact on patients’ quality of life [7]. However, to date, available therapeutic options aim solely at symptomatic improvement without being able to slow down, halt, or ideally reverse the underlying neurodegenerative process and, consequently, disease progression [8]. On the other hand, heterogeneity of PD in terms of age at symptoms’ onset, constellation of symptoms, progression, response to therapy, and genetic substrate may relate to distinct disease subtypes (endotypes), which may be linked to different underlying biological processes. Indeed, recent advances in PD genetics have shown that different pathogenetic mechanisms are involved in various degrees and combinations in different patients [9]. Biomarkers of several types (clinical, imaging, biochemical, genetic, and digital) may assist in better understanding the subjacent molecular pathways that drive PD development in each patient and guide individualized treatment [10]. Novel disease-modifying therapies that specifically target some, but not other, pathways are currently being developed in order to address this unmet need in the treatment of PD. Detection of patients at a prodromal or even a presymptomatic (preclinical) stage of PD, when neurodegeneration is at more initial stages, seems crucial, as this population is more likely to receive the full therapeutic benefit of the new targeted therapies [11,12].

PD diagnosis is primarily based on clinical assessment, with bradykinesia being the core symptom accompanied by rest tremor and/or rigidity [13]. Additional features may support (e.g., clear and positive response to levodopa) or exclude (e.g., ataxia) the diagnosis, or alert the treating physician to a potential alternative diagnosis of atypical or nondegenerative Parkinsonism (e.g., early falls, pyramidal signs), while nonmotor symptoms hold an essential role in identifying potential PD subtypes [14,15]. Early diagnosis of PD is challenging and significantly dependent on the clinician’s expertise, as to date there are no established diagnostic studies/tests that could efficiently substitute for clinical assessment [16].

Although not a prerequisite for PD diagnosis, resting tremor is a cardinal symptom in PD, with the vast majority of patients experiencing other type of tremors as well, such as postural or kinetic tremor [17,18]. Tremor is often the very first symptom noticed by PD patients or their caregivers, leading them to seek medical help. There are also indications that mild tremor may be present in the early or even prodromal phase of PD [19], but also in non-Parkinsonian carriers of selected PD-related genetic mutations [20], thus making it useful as a clinical biomarker of diagnostic or predictive value. Such motor phenomena may be so subtle that only instrumented motor assessments using portable devices can detect them, thus unraveling a population at risk.

In recent years, there has been growing interest in utilizing artificial intelligence (AI) techniques to develop noninvasive and accessible methods for early PD detection [21]. The advent of wearable devices, such as accelerometers embedded in smartphones or smartwatches, has provided a wealth of data which can be leveraged to extract meaningful information about an individual’s movement patterns and identify potential biomarkers for PD [22,23]. The use of machine learning (ML) and deep learning (DL) techniques that employ sensor signals (e.g., accelerometers, gyroscopes, pressure sensors) has garnered increased attention [24,25,26]. These wearable sensors offer a cost-effective solution for long-term monitoring under real-world conditions [27,28,29,30] in contrast to other analysis systems (such as video-based motion analysis) that require costly equipment and are restricted to indoor environments end experimental settings [31]. These findings indicate a promising future for monitoring and management of PD patients based on sensor signals and AI-related techniques.

Here, we present a novel approach for detecting PD in the wild using a bidirectional long short-term memory (Bi-LSTM) model with an attention mechanism. The key contribution of our work lies in the development of a robust and accurate model for PD detection in individuals with slight symptoms of tremor using accelerometer data collected in real-world settings. Unlike previous studies that primarily focused on controlled clinical environments, our approach aims to address the challenges of detecting PD in the wild, where data variability and noise are more prevalent. By leveraging the ubiquity of smartphones and smartwatches, our method allows for convenient and nonintrusive monitoring of individuals in their natural environments. Furthermore, our proposed architecture jointly captures temporal dependencies and focuses on relevant information within the accelerometer sequences. The use of Bi-LSTM layers enables the model to consider both past and future contexts, facilitating the extraction of discriminative features for classification of Parkinsonian symptoms severity. Additionally, the attention mechanism enhances the model’s ability to emphasize important segments of the input data, providing valuable insights into the underlying patterns contributing to PD detection.

Our paper follows a structured format to present the research findings in a clear and organized manner. It begins with an introduction which provides background information on PD and highlights the significance of detecting PD early, when symptoms are minor and clinically questionable. The Section 3 outlines the data collection process and describes the implementation of the Bi-LSTM model for classification. The Section 4 presents the experimental outcomes, including accuracy and loss curves, box plots, receiver operating characteristic (ROC) curves, and performance metrics. The Section 5 interprets the findings, explores the implications, and provides insights into the model’s performance, generalizability, and potential applications in early-stage PD detection. Finally, the Section 6 summarizes the key findings, highlighting the importance of the proposed model and its prospect towards PD detection when only subtle symptoms of tremor are present.

## 2. Literature Review: Deep Learning for Parkinson’s Disease Identification Based on Upper Limb Motion Data

### 2.1. Artificial Neural Networks

Artificial neural networks (ANNs) have revolutionized the field of healthcare and diagnosis, surpassing the efficacy and adaptability of conventional ML techniques [32]. ANNs have the potential to improve medical decision making, patient outcomes, and delivery of care. Inevitably, multilayer perceptron (MLP), but also deep neural network (DNN) models, have been developed additionally for diagnostic purposes related to PD. Authors in [33] proposed a novel mobile phone application system based on measuring the hand acceleration using a mobile phone accelerometer. The study utilized recordings from 21 PD patients and 21 HCs, extracting features through wavelet packet analysis to construct a 12-element feature vector. A neural network classifier achieved a 95% accuracy and a 90% kappa coefficient, indicating the reliable detection of PD. In another study, handwriting patterns were evaluated as potential biomarkers for PD [34]. The researchers collected 935 handwriting assignments from 55 PD patients and 94 HCs. Three feature sets were extracted: neuromotor, kinematic, and nonlinear dynamic. Among traditional ML classifiers, an MLP network was tested, achieving an accuracy of 97% and 78% for classifying between PD and young or elderly HCs, respectively.

Bazgir et al. [35] evaluated tremor using a supervised learning pattern recognition system. The study implemented a classifier block and neural network based on smartphone-recorded acceleration data. The results demonstrated a high accuracy of the neural network, showing a 91% correlation between the tremor item in the Unified Parkinson’s Disease Rating Scale (UPDRS) and acceleration values. This approach offered a potential method for evaluating tremor in PD patients. Finally, a study utilized signal processing techniques to examine the consistency between upper limb tremor scores in PD as assessed by the clinician and wearable devices [36]. Acceleration signals were collected during standard movements, and characteristic values were extracted using various domains. The neural network model showed higher accuracy compared to other models, achieving 95.3% accuracy for resting tremor scores and 91.2% accuracy for postural tremor scores. The combination of acceleration signals and displacement signals effectively simulated the clinician’s scores for tremors in the upper limb of PD patients.

### 2.2. Convolutional Neural Networks

Recently, convolutional neural networks (CNNs) have demonstrated remarkable capabilities in medical image analysis, disease detection, diagnosis, and prognosis [21,37]. These networks have the potential to significantly impact healthcare by contributing to early disease detection, treatment planning, and improved clinical outcomes. To this goal, several studies have explored the use of CNN in the context of PD and Parkinsonian tremor detection. Xing et al. [38] evaluated seven predictive models, including CNN, to differentiate between PD and essential tremor (ET) using demographics and upper limb tremor data. The CNN model, trained on tremor frequency and amplitude-related features, demonstrated modest predictive ability with accuracy above 78% and f1-score above 0.83. In another study, a CNN-based method for detecting PD hand tremor using wrist acceleration information collected from a wearable sensor device was proposed [39]. Their CNN model with nine layers outperformed conventional ML techniques in detecting PD hand tremor, reaching an accuracy of 97%.

The advantages of accelerometers and gyroscopes as wearable sensors on the wrists were exploited towards the development of TremorSense, a PD tremor detection system [40]. Researchers employed an eight-layer CNN model to classify rest, postural, and action tremor in PD, achieving an accuracy of over 94% in various evaluation scenarios. Moreover, in a recent study [41], scientists introduced and evaluated histograms of oriented gradients (HOGs) combined with ML and DL methods, including a 2D-CNN, for evaluation of tremor severity in PD patients. The HOG descriptor achieved high discriminating rates in identifying tremor patterns from handwritten drawings and the network achieved a 83% accuracy. Finally, a tremor assessment system utilizing a CNN to distinguish the severity of symptoms based on data collected from a wearable device was developed by Kim et al. [42]. Their CNN architecture demonstrated accurate tremor monitoring with 85% correct prediction rate. The aforementioned studies highlight the potential of CNN in PD and tremor detection, showing improved accuracy and performance compared to conventional techniques.

### 2.3. Long Short-Term Memory

Long short-term memory (LSTM) is a type of recurrent neural network (RNN) that has gained significant attention for its ability to model and analyze sequential data [43]. LSTM excels in capturing dependencies and patterns over long sequences, making it particularly well suited for time series and sequential data analysis. In healthcare, LSTM-based models have been successfully applied on various occasions, such as disease prediction, patient monitoring, and medical signal analysis [21,44]. These models leverage the memory capabilities of LSTM to capture temporal dynamics and contextual information, enabling accurate predictions and insights from healthcare data.

Taking the above into consideration, Thummikarat et al. [45] proposed the use of an LSTM network to aid physicians in PD diagnosis. Their study demonstrated an accuracy of 73% in identifying patients with PD at an early stage with the suggested architecture. In another study [46], RNN models were trained using figure drawing data represented as a time series of coordinates, angles, and pressure readings. The paper compared two recurrent network models, namely, LSTM and Echo State Networks, to analyze the advantages and limitations of each architecture. Finally, LSTM achieved an identification accuracy of 91% and an f1-score of 0.94.

Hssayeni et al. [47] used wearable devices and introduced two methods based on deep LSTM networks and gradient tree boosting to continuously monitor Parkinsonian tremor using gyroscope sensor signals recorded during various body movements. The evaluation of data acquired from 24 PD patients showed that the LSTM method exhibited relatively high correlation with the clinician’s assessment of tremor, as rated by UPDRS III subscores (r = 0.77, *p* < 0.0001) through subject-based leave-one-out cross-validation.

### 2.4. Hybrid Deep Learning Architectures

Through our literature search, we identified two studies that used sophisticated DL to distinguish between ET and Parkinsonian tremor. Initially, Hathaliya et al. [48] integrated gated recurrent unit (GRU) and LSTM algorithms. Specifically, accelerometer data were passed through the GRU model, and served as input into the LSTM model to improve its performance. The authors utilized a blockchain network to validate that the trained model’s testing accuracy was 74.1%. Additionally, in [49], a data-driven NeurDNet model was proposed for classifying tremor in PD and ET based on the kinematics of the hand. The network was trained on more than 90 h of hand motion signals, including 250 tremor evaluations from 81 patients, and its differential diagnostic accuracy of 95.55% exceeds its state-of-the-art equivalents.

Concerning the diagnosis of PD, researchers investigated the applicability of CNN and CNN-BLSTM models based on time series classification derived from pen-based signals [50]. In this context, the Multi-Modal Collection (PDMultiMC) containing recordings of online handwriting, vocal signals, and eye movements was created [51]. HandPDMultiMC, a subset of PDMultiMC, includes examples of handwriting from 42 participants (21 PD patients and 21 non-PD controls). When combined with synthetic data augmentation, CNN-BLSTM models trained with jittering and synthetic data augmentation provided the highest performance for early PD identification (97.62% accuracy), according to experimental results on the dataset.

### 2.5. Limitations in Previous Studies, Addressed in the Present Study

In comparison to previous studies, our work addresses several limitations and introduces novel contributions to the field of PD detection. Initially, the aforementioned studies focused on PD population with variable symptoms’ severity of the different PD symptoms. Nevertheless, in our study, and as explained in detail in Section 3.1, we selected PD patients experiencing only slight symptoms of tremor, thus focusing on a more homogeneous group of patients. Secondly, while previous studies primarily used controlled clinical environments, our approach specifically targets PD detection in real-world settings by leveraging data collected in the wild. This is crucial as it allows us to account for the variability and noise commonly encountered in everyday life, making our model more robust and applicable to diverse scenarios. Furthermore, our method leverages the widespread use of smartphones and smartwatches, enabling convenient and nonintrusive monitoring of individuals in their natural environments. This eliminates the need for specialized equipment or clinical visits, making it more accessible and cost-effective. Finally, our proposed architecture comprising LSTM layers and an attention mechanism (see Section 3.4) enables us to capture temporal dependencies effectively and highlight relevant information within the accelerometer sequences.

## 3. Methodology

### 3.1. Dataset Description

The dataset used in this study contains IMU signals captured in the wild via the accelerometer sensor embedded in modern smartphones for the purpose of detecting tremorous episodes related to PD [52]. A total of 31 PD patients and 14 HCs contributed accelerometer data from their personal devices over the course of several months. Automatic recording of triaxial acceleration values was performed whenever a phone call was made. The annotation in the dataset is based on UPDRS items 16, 20, and 21, from now on mentioned as updrs16, updrs20, and updrs21, respectively. updrs16 refers to the value related to tremor as described in item 16 of the UPDRS part II, as reported by the subject, and updrs20 and updrs21 to the values related to rest tremor and action/postural tremor, respectively, in the right and left hand as described in items 20 and 21 of the UPDRS part III, as reported by the attending neurologist. Finally, a pd_status of ‘1’ indicates that the subject is a PD patient, whereas a value of ‘0’ indicates that the subject is an HC.

For purposes of early diagnosis of PD based solely on the motor symptom of upper limb tremor, we used a subset of the initial dataset by keeping those subjects with updrs16 score ≤1 (i.e., 0 and 1). The reason we decided to use updrs16 and not updrs20 or updrs21 was to account for the participant’s perspective of their tremor. Thus, we only included individuals who may not have been aware of their tremor (even if it was objectively measured by the clinician in items 20 and/or 21 of UPDRS) or, if they had been, this tremor was perceived as minor without any intrusions in activities of everyday life (even if it was not captured in the clinician’s assessment using UPDRS). Due to the subtle symptoms, these individuals may not consider visiting a physician at this point, but the findings of portable technologies may alert them to seek medical help. We also thought that UPDRS item 16 may offer a more complete and concise estimation of patients’ tremor compared to UPDRS items 20 and 21. The latter items represent instant assessments that may not reflect the true nature of tremor, symptoms which, similarly to other PD symptoms, tend to fluctuate during the day due to several factors (stress, excitement, fatigue, posturing, etc.). Thus, the final utilized dataset comprises 22 PD patients and 14 HCs.

### 3.2. Data Preprocessing

To lessen the computing burden and network complexity of the system, the accelerometer data were downsampled to 100 Hz before being used in the training and evaluation stages. Notably, PT signals have an instructive spectral region up to 12 Hz in frequency [53]. Given that the Nyquist theorem states that a signal can be reconstructed in its entirety if it is sampled at a rate twice as fast as its highest informative frequency, we conclude that a sampling rate of 100 Hz is adequate for both minimizing the system’s computational load and preserving the signal’s spectral contents of interest. Last, but not least, a Butterworth filter with a 12 Hz cutoff was used to efficiently isolate and analyze the unique frequency range associated with PT [54].

### 3.3. Windowing and Feature Extraction

After completing the data preprocessing phase, we applied a windowing technique with each window size set to 100 data points (1 s) with 50% overlap. The purpose of introducing overlap between windows is to ensure that important temporal information in the time series is captured. It allows for smoother transitions between adjacent windows and reduces the chances of missing significant patterns or events that might occur near the edges of individual windows [55]. By defining the window size and overlap, we are specifying the granularity at which the time series data will be analyzed. Smaller window sizes with higher overlaps provide more detailed information but may increase computational complexity, while larger window sizes with lower overlaps offer a more aggregated view of the data but may potentially miss finer patterns [55].

For this work, thirty-three (33) time-, frequency-, and energy-related features were calculated for each window, as seen in Table 1.

### 3.4. Implementation of Bidirectional LSTM with Attention

LSTM is a type of RNN layer that can effectively process sequential data, such as time series or text [57]. The bidirectional variant of LSTM processes the input sequence in both forward and backward directions, allowing the model to capture dependencies in both past and future contexts. The use of a Bi-LSTM with attention model for classifying PD patients and HCs using accelerometer signals from smartphones during phone conversations is justified for several reasons [58].

Firstly, the Bi-LSTM architecture is well suited for capturing temporal dependencies in sequential data, which is essential for analyzing accelerometer signals collected over time. By processing the input sequence in both forward and backward directions, the model can effectively capture patterns and dependencies in both past and future context. This is particularly relevant in the context of PD classification, as the characteristics of movements associated with the disease may manifest in different ways over time. Additionally, the inclusion of an attention mechanism further enhances the model’s ability to focus on relevant parts of the input sequence. PD can exhibit subtle variations in movement patterns and intensity, and the attention mechanism allows the model to dynamically weigh the importance of different time steps in the sequence, emphasizing the most informative parts.

#### 3.4.1. Bidirectional LSTM Layer

As mentioned earlier, Bi-LSTM comprises an LSTM in both the forward and backwards directions. The memory controller in the LSTM is used to determine which information is forgotten and which is retained. It is implemented by three structures: the input gate, the forget gate, and the output gate. The structure of the unit is depicted in Figure 1. The operation process is as follows:(1)$$\begin{array}{cc} i_{t} & =\sigma (W_{xi}x_{t}+W_{hi}h_{t-1}+b_{i}) \end{array}$$(2)$$\begin{array}{cc} f_{t} & =\sigma (W_{xf}x_{t}+W_{hf}h_{t-1}+b_{t}) \end{array}$$(3)$$\begin{array}{cc} o_{t} & =\sigma (W_{xo}x_{t}+W_{ho}h_{t-1}+b_{o}) \end{array}$$(4)$$\begin{array}{cc} g_{t} & =\tanh(W_{xc}x_{t}+W_{hc}h_{t-1}+b_{c}) \end{array}$$(5)$$\begin{array}{cc} c_{t} & =f_{t}c_{t-1}+i_{t}g_{t} \end{array}$$(6)$$\begin{array}{cc} h_{t} & =o_{t}\tanh\left(c_{t}\right) \end{array}$$

The input gate (1) determines how much new information should be let through to update the cell state $c_{t}$ at time step *t*. It takes into account the current input $x_{t}$, the previous hidden state $h_{t-1}$, and bias $b_{i}$ using the weight matrices $W_{xi}$ and $W_{hi}$. The sigmoid function σ squashes the combined input, producing a value between 0 and 1, representing the gate’s openness (0 means closed, 1 means open). The forget gate (2) determines how much of the previous cell state $c_{t-1}$ should be forgotten at time step *t*. It takes into account the current input $x_{t}$, the previous hidden state $h_{t-1}$, and bias $b_{f}$ using the weight matrices $W_{xf}$ and $W_{hf}$. The sigmoid function σ outputs values between 0 and 1, indicating the amount of information to retain (1 means retain all, 0 means forget all). The output gate (3) determines how much of the cell state $c_{t}$ should be exposed as the hidden state $h_{t}$ at time step *t*. It takes into account the current input $x_{t}$, the previous hidden state $h_{t-1}$, and bias $b_{o}$ using the weight matrices $W_{xo}$ and $W_{ho}$. The sigmoid function σ produces values between 0 and 1, controlling the amount of information to be exposed in the hidden state. The new cell state $g_{t}$ (4) is updated by considering the current input $x_{t}$, the previous hidden state $h_{t-1}$, and bias $b_{c}$ using the weight matrices $W_{xc}$ and $W_{hc}$. The hyperbolic tangent function (tanh) maps the combined input to a value between −1 and 1, capturing new candidate information. The updated cell state $c_{t}$ (5) is obtained by combining the previous cell state $c_{t-1}$ with the new candidate information $g_{t}$. The forget gate $f_{t}$ determines how much of the previous cell state is forgotten, while the input gate $i_{t}$ determines how much of the new candidate information is incorporated. Finally, the hidden state $h_{t}$ is obtained by applying the output gate $o_{t}$ to the hyperbolic tangent of the updated cell state $c_{t}$. The output gate controls the exposure of the cell state information, while the hyperbolic tangent function applies a nonlinear transformation to capture relevant features.

The traditional LSTM network, however, can only learn in one direction, thereby disregarding the reverse information. In the bidirectional LSTM [59], nevertheless, the current input depends not only on the prior accelerometer value but also on the subsequent value. The combination of the two LSTM units thoroughly considers the temporal information preceding and following each acceleration value, as depicted in Figure 2.

$w_{i}\left(i=1,\ldots ,6\right)$ denotes the weight from one unit layer to another. $x_{t}$ is the feature vector, *h* means LSTM units of input feature sequence (…, $x_{t-1}$, $x_{t}$, $x_{t+1}$, …), *h*’ indicates LSTM units of input feature sequences (…, $x_{t+1}$, $x_{t}$, $x_{t-1}$,…), and $o_{t}$ is the corresponding output after the feature vector passes through the bidirectional LSTM network.

The operation process is expressed as follows:(7)$$\begin{array}{cc} h_{t} & =\sigma (w_{1}x_{t}+w_{2}h_{t-1}+b_{t}^{\left(1\right)}) \end{array}$$(8)$$\begin{array}{cc} h_{t}^{\prime } & =\sigma (w_{3}x_{t}+w_{5}h_{t-1}^{\prime }+b_{t}^{\left(2\right)}) \end{array}$$(9)$$\begin{array}{cc} o_{t}^{\prime } & =\tanh(w_{4}h_{t}+b_{t}^{\left(3\right)}) \end{array}$$(10)$$\begin{array}{cc} o_{t}^{\prime \prime } & =\tanh(w_{6}h_{t}^{\prime }+b_{t}^{\left(4\right)}) \end{array}$$(11)$$\begin{array}{cc} o_{t} & =0.5\times (o_{t}^{\prime }+o_{t}^{\prime \prime }) \end{array}$$
where $b_{t}^{\left(1\right)},b_{t}^{\left(2\right)},b_{t}^{\left(3\right)},b_{t}^{\left(4\right)}$ are the biases in the bidirectional LSTM network at time *t*. Moreover, $o_{t}^{\prime },o_{t}^{\prime \prime }$ are the results of two LSTM units dealing with the vector output at the corresponding time. As shown in (11), the output feature vector $o_{t}$ is the mean of the two vectors at the corresponding time. The vector is fed into the attention mechanism to learn the network’s weight.

#### 3.4.2. Attention Layer

By assigning attention weights to the hidden states, the attention layer provides a mechanism to selectively weigh the contributions of the forward and backward hidden states based on their relevance to the task at hand. The attention mechanism process is as follows:(12)$$\begin{array}{cc} e_{t} & =v_{a}^{T}\tanh\left(W_{a}\left[h_{t};h_{t}\right]+b_{a}\right) \end{array}$$(13)$$\begin{array}{cc} \alpha_{t} & =\mathrm{softmax}(e_{t}) \end{array}$$(14)$$\begin{array}{cc} c & =\sum_{t}\alpha_{t}\left[h_{t};h_{t}\right] \end{array}$$
where in (12), $h_{t}^{f}$ and $h_{t}^{b}$ are the forward and backward hidden states at time step *t*, respectively. $W_{a}$ is a weight matrix, $b_{a}$ is a bias term, and $v_{a}$ is a weight vector for calculating attention scores. The attention score $e_{t}$ is obtained by taking the dot product of $v_{a}$ with the hyperbolic tangent of the linear transformation of the concatenated hidden states. The attention scores $e_{t}$ in (13) are passed through the softmax function to obtain attention weights $\alpha_{t}$. The softmax function normalizes the attention scores across all time steps, ensuring that the weights sum up to 1. $\alpha_{t}$ represents the importance or weight assigned to each hidden state $h_{t}$ in the sequence. Finally, in (14), the context vector *c* is calculated by taking the weighted sum of the concatenated hidden states $\left[h_{t}^{f};h_{t}^{b}\right]$ across all time steps. The attention weights $\alpha_{t}$ determine the contribution of each hidden state to the context vector, and c represents a summary or representation of the input sequence, capturing the most relevant information.

#### 3.4.3. Network Architecture

The proposed model, which is presented in Figure 3, consists of multiple bidirectional LSTM layers to capture sequential patterns in the input accelerometer data. Dropout layers are added for regularization, and an attention layer helps the model focus on relevant information. Finally, a flatten layer and dense layer with a sigmoid activation function produce the binary classification output.

The hyperparameters of various layers of the presented model are shown in Table 2. Notably, the final model consists of a total of 346,349 parameters.

#### 3.4.4. Experimental Setup and Implementation

Python version 3.9.7 installed on a personal computer with a GTX GeForce 750 Ti GPU, Intel(R) Core(TM) i7-6700 CPU at 3.40 GHz clock speed, and 32 GB of RAM was used for the implementation and experimentation. TensorFlow-GPU version 2.5.0 with the frontend of Keras-GPU were used for developing, training, and evaluating the DL model. During training, the model’s loss was calculated using a binary cross-entropy loss function, and the model’s weights were updated using an Adam optimizer with an initial learning rate of $10^{-4}$. In addition, 128 minibatch sizes were used to train the suggested model.

## 4. Results

Regarding our experimental procedure and the classification stage, we split the data into training and testing sets, with the number of test data samples being 30% of the total number of examples [60]. To assess the performance of our model, we employed a fivefold cross-validation (5-CV) technique, ensuring a robust evaluation. Additionally, hyperparameter optimization was performed using the GridSearch method, allowing us to find the optimal values for the model’s hyperparameters. Furthermore, we introduced an early stopping callback function with a patience level set to 3, which prevented overfitting and improved the model’s generalization capability.

### 4.1. Training and Validation

The training and validation progress of the proposed model was visualized using accuracy and loss curves for each fold of the 5-CV. The accuracy curves depict the performance of the model in correctly classifying PD and HC samples, while the loss curves represent the convergence of the model during training.

Figure 4 illustrates the accuracy curves for the training and validation sets across the five folds. It can be observed that the model achieves high accuracy on both the training and validation sets, indicating its ability to effectively learn the underlying patterns in the accelerometer data. The accuracy steadily increases during the initial epochs and eventually plateaus, demonstrating the convergence of the model. Notably, the validation accuracy closely follows the training accuracy, suggesting that the model generalizes well to unseen data.

In Figure 5, the loss curves for the training and validation sets are presented. The curves depict the reduction of the loss function over the training epochs. As expected, the loss decreases rapidly during the initial epochs and gradually stabilizes as the model converges. The loss curves for the training and validation sets exhibit a similar pattern, indicating that the model is not overfitting to the training data and is able to generalize well to new samples.

Furthermore, we visualized the distribution of performance metrics across the different folds and epochs using box plots, as observed in Figure 6. These plots provide valuable insights into the variability and consistency of our model’s performance during training and validation. The first set of box plots represents the training and validation loss across different folds at each epoch. The vertical axis denotes the loss values, while the horizontal axis represents the folds. The boxes in the plots indicate the interquartile range (IQR) of the loss distribution, with the median value represented by the horizontal line inside the box. The whiskers extending from the boxes indicate the minimum and maximum values, excluding outliers.

Similarly, we created box plots to visualize the training and validation accuracy across different folds at each epoch. These plots showcase the distribution of accuracy values achieved by our model during training and validation. Once again, the boxes represent the IQR, the horizontal line inside the box denotes the median accuracy, and the whiskers display the minimum and maximum values, excluding outliers.

Analyzing these box plots provides a comprehensive understanding of our model’s performance characteristics. We can observe the consistency of our model’s performance across different folds, with minimal variability in both loss and accuracy. This consistency indicates the robustness of our proposed model in effectively capturing and generalizing patterns in the data.

To further assess the discriminative power and performance of our proposed model, we plotted the ROC curves and calculated the corresponding area under the curve (AUC) for each fold, as presented in Figure 7. The curves show the model’s performance at different classification thresholds, indicating the balance between correctly identifying PD patients and minimizing false positive predictions.

Analyzing the ROC curves and AUC values across the five folds, we observed consistent and high performance across all iterations. The AUC values for each fold were consistently high, indicating that the Bi-LSTM with attention model achieved effective discrimination between PD patients and HCs. These results highlight the robustness and effectiveness of our model in accurately classifying PD patients with minor tremor symptoms.

### 4.2. Performance Evaluation on the Test Set

The performance of our network model was evaluated using several metrics, including accuracy, precision, recall, specificity, and f1-score. These metrics provide insights into different aspects of the model’s performance. The average performance metrics obtained from the 5-CV are presented in Table 3.

The results demonstrate the effectiveness of our proposed model in accurately classifying the accelerometer data into PD patients and HCs. The high mean accuracy of 0.98 indicates that our model achieved a high overall classification accuracy across all folds. The mean precision of 0.99 indicates a low rate of false positives, emphasizing the model’s ability to correctly identify positive instances. The mean recall of 0.98 suggests that the model effectively captures the majority of positive instances. The mean specificity of 0.96 indicates a high rate of correctly identifying negative instances. Lastly, the mean f1-score of 0.98 represents a balanced measure of precision and recall, indicating the model’s strong overall performance.

Additionally, to provide a comprehensive comparison with the relevant literature, Table 3 presents the performance metrics of our proposed model alongside the results reported in other research works. Notably, our Bi-LSTM with attention model achieved superior performance compared to previous studies. The comparison underscores the effectiveness of our proposed model in accurately differentiating PD patients from healthy individuals, particularly when minor symptoms are present, highlighting its significance and potential as a valuable tool for early PD detection and monitoring when presenting symptoms may be subtle. Importantly, our model’s success can be attributed to its ability to analyze accelerometer data collected in real-world settings, capturing the true complexities and variabilities of individuals’ everyday movements. This feature enhances the generalizability and practicality of our approach, making it suitable for real-life applications and paving the way for widespread adoption in the field of PD research and healthcare.

### 4.3. An Additional Experiment with a Subset of the Dataset

We conducted an additional experiment for the purpose of early diagnosis of PD based solely on the motor symptom of upper limb tremor. This time, we utilized an even smaller sample of the initial dataset by keeping those subjects with only zero (0) updrs16 score. We, thus, wanted to address this group of patients who either do not experience any symptoms of tremor (e.g., akinetic type of PD) or if they did (updrs20/21 score of at least 1), they were not aware of it. In this case, we ended up with a much smaller dataset which comprised 8 PD patients and 13 HCs. For this particular experiment, we followed the same data processing, feature engineering, training, validation, and testing procedure as described in the previous sections. Our Bi-LSTM with attention network once more yielded encouraging results, with the metrics of accuracy, precision, recall, specificity, and f1-score being 0.97, 0.98, 0.96, 0.96, and 0.96, respectively.

## 5. Discussion

The results of this study demonstrate the effectiveness of our proposed Bi-LSTM with attention model in accurately discriminating PD patients from HCs based on accelerometer data when individuals experience slight or no tremor (updrs16 score ≤1), irrespective of the clinician’s objective assessment. The model achieved high accuracy, precision, recall, specificity, and f1-score, indicating its robustness and potential for detection and monitoring of PD when only minor motor symptoms of tremor are present, a concept which may apply in patients with early or even prodromal PD.

One key advantage of our model is its ability to efficiently analyze accelerometer data collected in real-world settings. By capturing the true complexities and variabilities of individuals’ everyday movements, our model enhances the generalizability and practicality of methods of PD detection. This capability is particularly crucial for early-stage PD detection, as it enables the model to handle diverse and dynamic movement patterns commonly observed in real-life scenarios.

The high accuracy achieved by our model surpasses the performance of previous studies in PD classification using different types of data, such as typing, writing, and hand poses. This superiority can be attributed to the effectiveness of the Bi-LSTM architecture combined with the attention mechanism. The Bi-LSTM allows the model to capture temporal dependencies in the accelerometer data, while the attention mechanism focuses on relevant segments of the input, enhancing the model’s ability to extract discriminative features.

The results of the fivefold cross-validation demonstrate the model’s stability and consistency in performance across different folds. The close alignment between training and validation accuracy curves suggests that our model generalizes well to unseen data, indicating its robustness and ability to capture the underlying patterns in the accelerometer data. Moreover, the analysis of ROC curves and AUC values further supports the discriminative power of our model. The well-separated ROC curves and consistently high AUC values across the five folds indicate the model’s ability to distinguish between PD patients and HCs with high sensitivity and specificity. This implies that our model can effectively balance between correctly identifying PD patients and minimizing false positive predictions.

Comparing our results with previous works, our proposed model outperformed existing studies in PD classification. Specifically, in the work by [45], an LSTM model was applied to classify PD patients and HCs based on typing data, achieving an accuracy of 0.73. Researchers in [46] utilized a two-stacked LSTM model for PD detection during writing tasks, resulting in an accuracy of 0.91. Moreover, a CNN-BiLSTM model was employed in [50] for classifying PD based on writing data, achieving an accuracy of 0.98. Scientists in [33] used a DNN model for PD classification using hand pose data, obtaining an accuracy of 0.95. Comparatively, our proposed model exhibited superiority in terms of accuracy, precision, recall, specificity, and f1-score.

While our proposed model outperforms previous works, several limitations need to be acknowledged. Firstly, the participants’ clinical and accelerometer data were derived from an open access public dataset comprising PD patients with different characteristics in terms of disease severity/duration and levodopa equivalent dose [61]. Although all included patients had comparative tremor scores in the relevant UPDRS items (≤1 in updrs16 and ≤2 in updrs20/updrs21), patients may have not been similar regarding their disease severity, which is defined by other symptoms as well, like bradykinesia or gait impairment. It is, thus, not clear whether the participants’ differentiation between PD patients and controls was based solely on the detection of slight symptoms of tremor, potentially macroscopically unperceived by human senses (both of the patient and of the examiner), or due to coexistent motor features, such as bradykinesia. Indeed, accelerometers capture motor performance collectively, based on the recorded acceleration of movements. In this particular experiment, data were collected for brief time periods while individuals were talking on their smartphone and their upper limb was relatively stable. Although a true state of immobility cannot be realistically achieved, it is considered less likely to capture symptoms of bradykinesia when no particular movements are actively performed (e.g., after a command). Moreover, bradykinesia, in contrast to rest or postural tremor, is usually detected from accelerometers when upper or lower limbs are actively moving [62,63]. We could, thus, argue that the selected group of patients was rather homogeneous from the scope of tremor characteristics.

The high heterogeneity of PD cannot be captured by a single symptom though, and, in our case, tremor-free PD patients of the akinetic type are bound to be missed by such a detection system. In order to address this concern, we performed a further analysis of a subset of PD patients who reported no symptoms of tremor (0 in updrs16). Interestingly, our model achieved high levels of PD detection in this subgroup of individuals as well, highlighting that the built-in accelerometer may be able to capture particular patterns of motor impairment that patients are not aware of. Although the results of this latter subanalysis should be received with skepticism due to the small sample size which can possibly introduce bias, high variance, limited complexity, and generalization, they pave the way for future studies. Expanding the dataset size and diversity could enhance the generalizability of the model. Moreover, tremor tends to improve with dopaminergic therapy and occasionally declines as the disease progresses and symptoms of rigidity and bradykinesia predominate. Thus, low tremor scores may easily be encountered both in the early and advanced stage of PD; this cannot be inferred by UPDRS, which is mostly a descriptive tool. A dataset of drug-naïve, early-stage PD patients, or even at-risk individuals in the preclinical or prodromal stage of PD would give more focused and robust results considering early PD detection.

Additionally, tremor is a common symptom, which may be associated with numerous other neurological (e.g., ET) and systemic disorders (e.g., thyroid dysfunction), and may also arise as a drug-related adverse event (e.g., valproic acid) or even be present in healthy individuals (physiological tremor) [64]. Including participants with such characteristics in future datasets would also be essential. Incorporating other types of sensor data, such as voice recordings or gait analysis, could provide a more comprehensive and multimodal approach to PD detection. Finally, exploring the interpretability of the model’s predictions and investigating the specific features learned by the attention mechanism could also provide valuable insights into the underlying mechanisms of PD.

## 6. Conclusions

In this study, we proposed a novel approach for detection of PD using accelerometer data collected during phone conversations from subjects demonstrating minor tremor of the upper limb, which might have gone undetected. The results of our experiments showcase the robustness and effectiveness of our model in differentiating PD patients from HCs. Our findings may have significant implications for the field of PD research and healthcare. The ability to accurately detect subtle PD symptoms, such as tremor, could potentially facilitate the identification of PD patients in the early stage of the disease and prompt individuals at risk to seek medical help and proper consultation, and eventually lead to its effective treatment. Such a possibility could also lead to effective patients’ selection for clinical trials of new targeted, disease-modifying precision-medicine therapies and for biomarkers development. Our proposed model provides a valuable contribution to this end, as mobile-based applications capturing data in the wild can revolutionize screening procedures for outpatients. Future research directions include expanding the dataset size and diversity, incorporating other sensor modalities, such as voice recordings or gait analysis, and investigating the interpretability of the model’s predictions. These efforts may further enhance our understanding of the underlying mechanisms of PD.

## Figures and Tables

**Figure 1 sensors-23-07850-f001:**
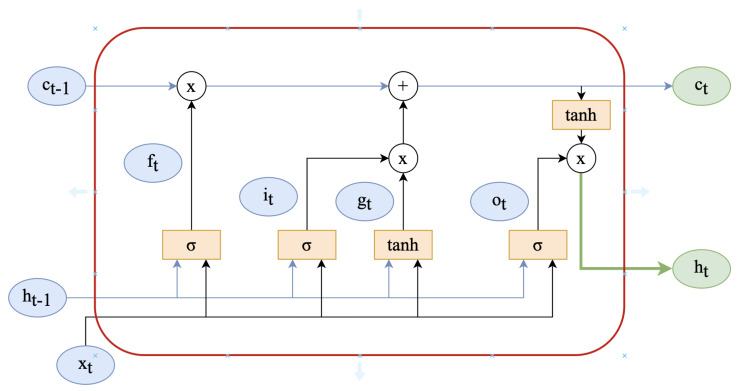
LSTM cell internal architecture.

**Figure 2 sensors-23-07850-f002:**
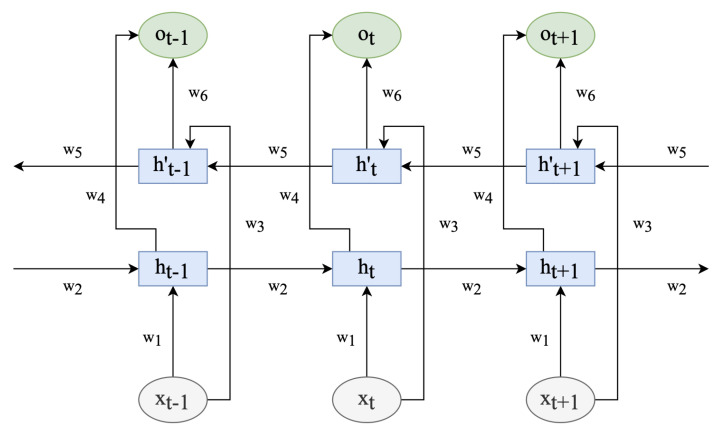
Bidirectional LSTM network model.

**Figure 3 sensors-23-07850-f003:**
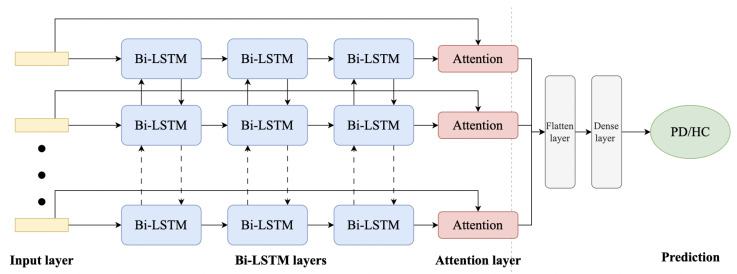
Proposed Bi-LSTM with attention network architecture.

**Figure 4 sensors-23-07850-f004:**
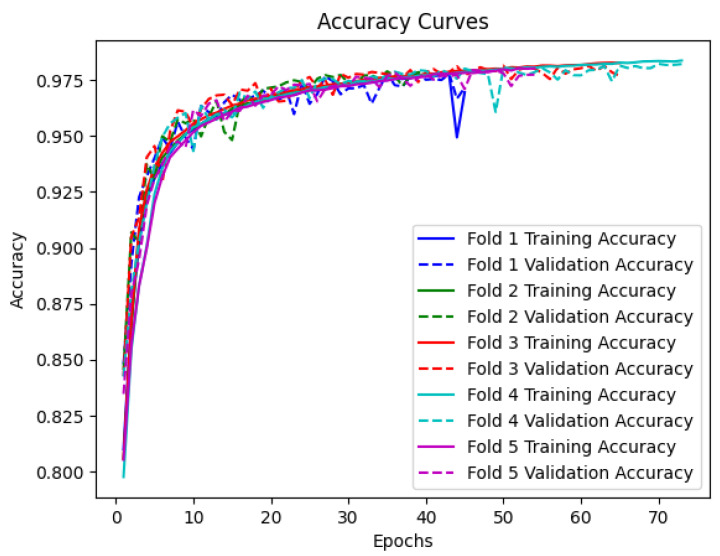
Training and validation accuracy curves for every epoch for BiLSTM with attention model.

**Figure 5 sensors-23-07850-f005:**
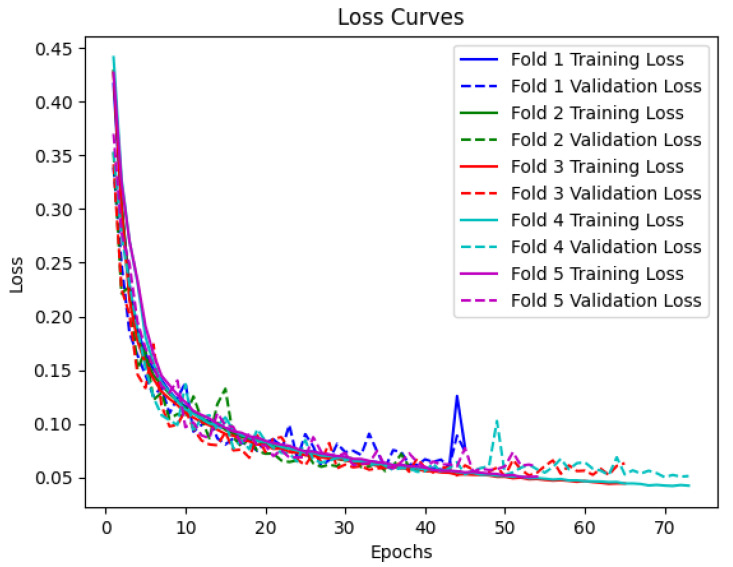
Training and validation loss curves for every epoch for BiLSTM with attention model.

**Figure 6 sensors-23-07850-f006:**
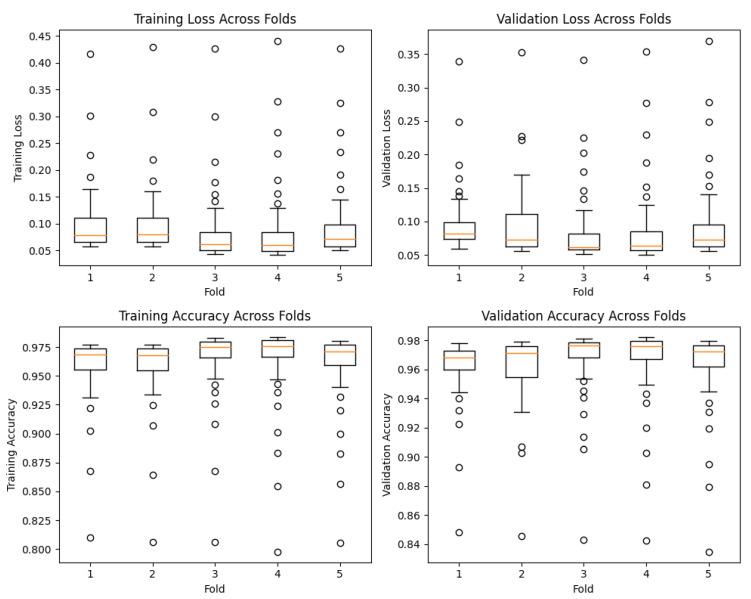
Box plots showing the distribution of training loss, validation loss, training accuracy, and validation accuracy across different folds at each epoch.

**Figure 7 sensors-23-07850-f007:**
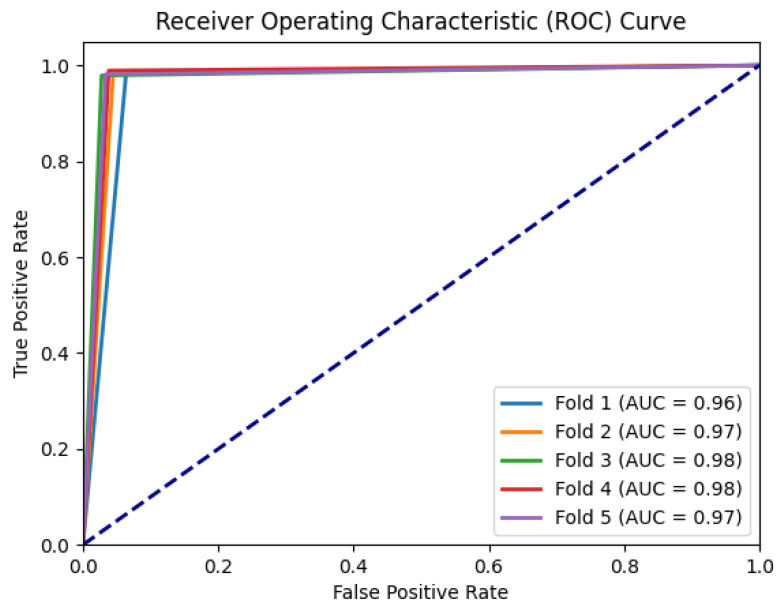
ROC curves for each fold for BiLSTM with attention model.

**Table 1 sensors-23-07850-t001:** Time-, frequency-, and energy-related features calculated per window (x) with size N = 100.

Feature	Description	Formula
Maximum	The maximum value in the signal (per axis).	$\max\;(x_{1},x_{2},\ldots ,x_{N})$
Minimum	The minimum value in the signal (per axis).	$\min\;(x_{1},x_{2},\ldots ,x_{N})$
Mean	The average value of the signal (per axis).	$\frac{1}{N}\sum_{i=1}^{N}x_{i}$
Standard deviation (std)	A measure of the dispersion or spread of the signal values around the mean (per axis).	$\sqrt{\frac{1}{N}\sum_{i=1}^{N}{\left(x_{i}-\mathrm{mean}\right)}^{2}}$
Kurtosis	A measure of the peakedness or flatness of the signal’s distribution (per axis).	$=\frac{\frac{1}{N}\sum_{i=1}^{N}{\left(x_{i}-\mathrm{mean}\right)}^{4}}{{\left(\frac{1}{N}\sum_{i=1}^{N}{\left(x_{i}-\mathrm{mean}\right)}^{2}\right)}^{2}}-3$
Zero Crossing Rate	The rate at which the signal changes its sign.	$\frac{1}{N-1}\sum_{i=1}^{N-1}\left(\left\{\begin{array}{cc} 1 & \mathrm{if}\;x_{i}\cdot x_{i+1}<0\\ 0 & \mathrm{otherwise} \end{array}\right.\right)$
Skewness	A measure of the asymmetry of the signal’s distribution (per axis).	$\frac{\frac{1}{N}\sum_{i=1}^{N}{\left(x_{i}-\mathrm{mean}\right)}^{3}}{{\left(\frac{1}{N}\sum_{i=1}^{N}{\left(x_{i}-\mathrm{mean}\right)}^{2}\right)}^{\frac{3}{2}}}$
Correlation	The correlation coefficients between different signal components or dimensions (for xy, xz, yz axes).	$\left(\begin{array}{ccc} {\mathrm{corr}}_{xy} & {\mathrm{corr}}_{xz} & {\mathrm{corr}}_{yz} \end{array}\right)$
Maximum PSD	The maximum power spectral density value in the signal.	max(psd)
Average PSD	The average power spectral density value in the signal.	$\frac{1}{N}\sum_{i=1}^{N}{\mathrm{psd}}_{i}$
Standard Deviation of PSD	A measure of the variation or spread of the power spectral density values.	$\sqrt{\frac{1}{N}\sum_{i=1}^{N}{\left({\mathrm{psd}}_{i}-\mathrm{avg}\underline{\;}\mathrm{psd}\right)}^{2}}$
Spectral Centroid	The center of mass of the power spectral density distribution, representing the average frequency content of the signal.	$\frac{\sum_{i=1}^{N}f_{i}\cdot {\mathrm{psd}}_{i}}{\sum_{i=1}^{N}{\mathrm{psd}}_{i}}$
Spectral Rolloff	The frequency below which a specified percentage of the total power of the signal is contained.	$0.85\cdot \sum_{i=1}^{N}{\mathrm{psd}}_{i}$
Spectral Flatness	A measure of the tonality or noisiness of the signal.	$\frac{\exp\left(\frac{1}{N}\sum_{i=1}^{N}\log\left({\mathrm{psd}}_{i}\right)\right)}{\frac{1}{N}\sum_{i=1}^{N}{\mathrm{psd}}_{i}}$
Spectral Skewness	A measure of the asymmetry of the power spectral density distribution around its centroid.	$\frac{1}{N}\sum_{i=1}^{N}{\left(\frac{f_{i}-\mathrm{spectral}\_\mathrm{centroid}}{\mathrm{std}\_\mathrm{psd}}\right)}^{3}$
Spectral Kurtosis	A measure of the peakedness or flatness of the power spectral density distribution around its centroid.	$\frac{1}{N}\sum_{i=1}^{N}{\left(\frac{f_{i}-\mathrm{spectral}\_\mathrm{centroid}}{\mathrm{std}\_\mathrm{psd}}\right)}^{4}-3$
Entropy	A measure of the randomness or unpredictability of the signal, calculated using Shannon’s entropy formula [56].	$-\sum_{i=1}^{N}p_{i}\log_{2}\left(p_{i}\right)$
Total Energy	The total energy or power in the signal, calculated as the sum of the squared values.	$\sum_{i=1}^{N}\left(x_{i}^{2}+y_{i}^{2}+z_{i}^{2}\right)$
Signal Magnitude Area	The sum of the absolute values of the signal.	$\sum_{i=1}^{N}\left(\left|x_{i}\right|+\left|y_{i}\right|+\left|z_{i}\right|\right)$

PSD: power spectral density

**Table 2 sensors-23-07850-t002:** Hyperparameters of the proposed architecture.

Layer	Neurons	Output Shape	Parameters	Activation
Input layer	33	(-, 33, 1)	0	-
Bi-LSTM layer 1	32	(-, 33, 64)	8704	-
Dropout 1	-	(-, 33, 64)	0	-
Bi-LSTM layer 2	64	(-, 33, 128)	66,048	-
Dropout 2	-	(-, 33, 128)	0	-
Bi-LSTM layer 3	128	(-, 33, 256)	263,168	-
Attention layer	-	(-, 33, 256)	0	-
Dropout 3	-	(-, 33, 256)	0	-
Flatten layer	-	(-, 8448)	0	-
Dense layer	1	(-, 1)	8449	Sigmoid

**Table 3 sensors-23-07850-t003:** Performance metrics for classification of PD patients and HCs, and comparison with therelevant literature.

Model	Experiment	Accuracy	Precision	Recall	Specificity	F1-Score
LSTM [45]	Typing	0.73				
Two-Stacked LSTM [46]	Writing	0.91	-	1.00	0.65	0.94
CNN-BiLSTM [50]	Writing	0.98	-	0.95	1.00	
DNN [33]	Hand poses	0.95	-	-	-	-
BiLSTM + attention (proposed model)	Talking on the phone	0.98	0.99	0.98	0.96	0.98

## Data Availability

The data presented in this study are openly available in Zenodo.org at https://doi.org/10.5281/zenodo.7273759 (accessed on 14 August 2023).

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
