# Peer review of "Detecting Minor Symptoms of Parkinson’s Disease in the Wild Using Bi-LSTM with Attention Mechanism"

_sensors, 2023, doi:10.3390/s23187850_

Round 1

Reviewer 1 Report

Reject and do not reconsider, I have some major concerns:

1-A state-of-the-art model: Bidirectional Long Short-Term Memory (Bi-LSTM) model with attention mechanism, used for classification. What is your contributions?

2-In the abstract, it is written, "appropriate mobile-based applications may be a practical tool in real-life settings to alert individuals at risk to seek medical assistance", why you have not created an app in this research? 

3-There are alot of research articles already published on Parkinson’s Disease, such as DOI:https://doi.org/10.1080/00207721.2012.724114 .This given paper is published in 2012, almost 11 years before. Also, it achieved 100% accuracy, your accuracy is also low.

4-please proofread the manuscript. I found many typos and grammatical errors. There is a need for extensive English revision.

5-In summary, the authors only try to replicate results, which are already achieved by machine learning.

Minor editing of English language required

Author Response

Please see the attached PDF file.

Reviewer 2 Report

The research is good and interesting, and the following points should be addressed:

1. The study's methodology must be drawn in full so that it is easy for readers to follow and understand the work.

2. The following words “May, May be, can and can be” should be deleted and replaced with an appropriate one.

3. The results should be explained further.

4. Does the data set contain missing and outliers? If yes, how was it processed?

5. Was the data set subjected to the data standardization method?

6. How many features of the data set and were they fully used during the classification?

7. Were techniques used to give the features priority by correlating each feature to the target feature?

8. What are the future works and what are the limitations the authors face?

 Minor editing of English language required.

Author Response

Please see the attached PDF file.

Reviewer 3 Report

accept with minor revision.

Author Response

Please see the attached PDF file.

Round 2

Reviewer 1 Report

Accepted as it is.

Reviewer 2 Report

Accept in this form

Good